# Chemical Transformations in Proto-Cytoplasmic Media. Phosphorus Coupling in the Silica Hydrogel Phase

**DOI:** 10.3390/life7040045

**Published:** 2017-11-19

**Authors:** Ian B. Gorrell, Timothy W. Henderson, Kamal Albdeery, Philip M. Savage, Terence P. Kee

**Affiliations:** School of Chemistry, University of Leeds, Woodhouse Lane, Leeds LS2 9JT, UK; I.B.Gorrell@leeds.ac.uk (I.B.G.); timhenderson879@gmail.com (T.W.H.); cmka@leeds.ac.uk (K.A.); sav1415@hotmail.co.uk (P.M.S.)

**Keywords:** silica, hydrogel, phosphorus, coupling, prebiotic

## Abstract

It has been proposed that prebiotic chemical studies on the emergence of primitive life would be most relevant when performed in a hydrogel, rather than an aqueous, environment. In this paper we describe the ambient temperature coupling of phosphorus oxyacids [Pi] mediated by Fe(II) under aerobic conditions within a silica hydrogel (SHG) environment. We have chosen to examine SHGs as they have considerable geological precedence as key phases in silicification *en route* to rock formation. Following a description of the preparation and characterization studies on our SHG formulations, coupling experiments between Pi species are described across multiple permutations of (i) Pi compound; (ii) gel formulation; (iii) metal salt additive; and (iv) pH-modifying agent. The results suggest that successful Pi coupling, indicated by observation of pyrophosphate [PPi(V)] via ^31^P-NMR spectroscopy, takes place when the following components are present: (i) a mixture of mixture of Pi(III) and Pi(V) or pure PPi(III– V); (ii) Fe(II); (iii) acetic or formic acid (not hydrochloric acid); (iv) aerobic conditions or the presence of H_2_O_2_ as an oxidant; and (v) the presence of a gel system. On the basis of these, and aqueous control reactions, we suggest mechanistic possibilities.

## 1. Introduction

Within the field of abiogenesis, significant advances have been made in many chemical, prebiotic directions, including synthetic routes to peptides [1,2,3,4,5,6,7,8], nucleic acid monomers [9,10,11,12] and subsequent oligomerization [13,14,15], proto-cellular assemblies [16,17,18,19,20] and proto-bioenergetic systems [21,22,23,24,25]. In all of these processes, one is manipulating systems that lie far from equilibrium in either a structural and/or dynamic sense [25]. In many cases, success requires control over local water activity [26] via, for example, wetting–drying cycles [7], non-aqueous media [27], and interaction with mineral surfaces [28] or frozen ice media [29].

It is envisaged that local environments in which the most primitive living systems may have emerged would have possessed properties that permitted, at least, the following functions: (i) molecular ingress and egress; (ii) water activity modulation; (iii) catalysis of chemical processes; (iv) a barrier between internal and external matrix; and (v) the ability to hold charge and concentration gradients [30]. However, observations and suggestions made by Trevors and Pollack over several years [31,32,33,34,35,36], in an attempt to provide a new perspective on the above boundary conditions, provide an interesting angle on origin of life studies. If one is trying to understand the emergence of the simplest form of biological life on Earth, that form being commonly referred to as LUCA, the Last Universal Common Ancestor [37], then it is arguably logical to envisage that the internal environment of LUCA might closely resemble the internal milieu of simple bacteria and archea. A cellular interior is not simply aqueous, nor is it simply a salt-water solution. It is best viewed as a hydrogel, a molecularly crowded, water-filled, cytoplasmic environment which is supported by some polymer or polymer-composite matrix with the ability to alter molecular diffusion, polymer structure and potentially, chemical reactivity [31,32,33,34,35,36]. Furthermore, a hydrogel is capable of maintaining physical integrity without the need for a membrane, providing a natural ability to exclude solutes according to their size. Thus, valuable charge and ion gradients, capable of being exploited for the transduction of energy [38], can be maintained naturally across a gel material without rapid dissipation. Moreover, recognition of the importance of molecular crowding within cellular gel-type environments as influencing aqueous structure and properties on the micro-scale [39,40,41] is a well-established phenomenon in biology. Whilst there has been clear recognition of the importance of a molecularly crowded environment to biological processes, attempts to translate that into an early Earth, prebiotic environment without compartmentalization have led to mixed results. For example, whilst two-phase and coacervate systems have long been proposed as effective membrane-free protocellular models [42,43,44], Szostak and co-workers found that they did not always provide effective segregation of molecular components [45].

We recognize the potential significance of a gel environment, especially in terms of selectively concentrating molecular building blocks [45], modifying water activity [35], influencing molecular diffusivity and hence residence time within the gel [34], manipulating molecular conformation of macromolecules and affecting the course of chemical reactions either thermodynamically in terms of product speciation or kinetics [39]. The question for us, then, was this: what form of hydrogel environment should one be looking at in order to be as prebiotically relevant as possible? Our response to this question was to consider geologically established environments based on silica hydrogels (SHGs) [46,47,48].

We have, therefore, embarked on a series of studies designed to explore differences between the outcomes of certain chemical processes when performed in the silica hydrogel phase as opposed to the aqueous phase. In doing so we have not attempted to mimic a specific, known geological silica gel deposit, but to focus on how certain processes may be influenced when the phase is changed from aqueous to hydrogel In this paper we explore the coupling of phosphorus (P) oxyacids to afford condensed phosphate oxyacids, specifically pyrophosphate PPi(V). Condensed P-species are ubiquitous energy currency molecules in contemporary biochemistry, for example the nucleotide triphosphates (Figure 1). Various, chemically less complex condensed P-species have been proposed to have been plausibly available to drive prebiotic chemistry. The latter include pyrophosphate [PPi(V)] [49,50,51], cyclo-trimetaphosphate (c-TMP) [52,53,54] and pyrophosphite [PPi(III)] (all of which contain the P-O-P linkage; Figure 1) [55,56,57]. However, the coupling of mono-phosphorus species to condensed P-compounds is usually observed to take place under conditions which are either thermally forcing [58], wet–dry cycles [59] or in the presence of activating agents [60]. In this paper, we demonstrate that aqueous phase mixtures of Pi(V) (H_2_PO_4_^−^) and Pi(III) (H_2_PO_3_^−^) which do not ordinarily undergo condensation to afford PPi(V) under ambient temperature conditions, do indeed do so when incubated within a silica hydrogel environment. Moreover, we examine possible mechanisms for such coupling and provide a plausible *modus operandi* for the gel-phase composite to undergo condensation. We have chosen to examine the role of Pi(III) in this study as it is the P-compound with which we have worked most often. However, reduced P-compounds are rare in contemporary biology and alternative methods for mobilizing, activating and coupling P-species have been proposed, see [49,50,51,52,53,54,60] and references therein.

## 2. Materials and Methods

Monosodium orthophosphate (NaH_2_PO_4_) and monosodium phosphite (NaH_2_PO_3_) were either commercial samples or were prepared by treating appropriate solutions of orthophosphoric acid or orthophosphorous acids (purchased from Sigma-Aldrich, Gillingham, UK) with equimolar amounts of sodium hydroxide (from Fisher Scientific, Loughborough, UK) in purified water at ambient temperature. Isohypophosphate [PPi(III– V)] was prepared according to the published method [56] and was composed of a mixture comprising [PPi(III– V)]-[Pi(III)/Pi(V)] in the relative proportions: 44–56% respectively (Appendix A). All water used herein was ultra-pure deionized water prepared using the Purite Select Analyst deionization system. Sodium silicate solution was commercial (from Merck KGaA, Darmstadt, Germany) and contained (≤27% SiO_2_ and ≤10% NaOH). Hydrochloric acid, silica gel (GF_254_), hydrogen peroxide (H_2_O_2_), glacial acetic acid, formic acid and metal salts (FeCl_2_·4H_2_O; FeSO_4_·7H_2_O; Fe(NO_3_)_3_·9H_2_O; CuSO_4_·5H_2_O; MgCl_2_) were commercial samples (from Fisher, Riedel de Haën, Seelze, Germany; Aldrich, Fluka, Buchs, Switzerland or Merck KGaA, Darmstadt, Germany) and were used as received.

### 2.1. Analytical Methods

Critical-point drying (CPD) was performed using a Polaron 3100 CPD instrument. Samples of silica hydrogel (prepared according to the procedure described in Section 2.2 below) were prepared inside 3.5 cm length (1.0 cm OD neck) plastic cuvettes, the lids of which had been punctured with 3–4 small needle holes. The gels were then incubated with acetone to remove the water and then transferred to fresh acetone in the CPD instruments boat. This was loaded into the CPD, where the acetone was replaced with liquid CO_2_ with multiple flushes over the space of 1 h. Subsequently, the system was heated to 35 °C via the attached thermo-regulator water system for ca 30 min, until the critical pressure of 1200 psi had been reached. At this point, the system/CO_2_ gas was vented to allow the dried samples to be removed. Sample matrices dried thus were analyzed via scanning electron microscopy as outlined below. Samples were dialyzed prior to SEM analysis by incubation in sealed dialysis tubing in ultra-pure deionized water for a period of four days.

Scanning electron microscopy was performed in the School of Chemistry, using an FEI Nova NanoSEM 450 instrument operating at 3 kV or 18 kV for EDX. Samples were iridium-coated prior to viewing and were inserted into the sample chamber under high vacuum (6.3 × 10^−6^ Torr). The image magnification varied from 100 nm to 1 µm with high-resolution images collected in the secondary electron mode. Element analysis (EDX) was performed using AMETEK software TEME V 3.4 and surface-area measurements were made using a Micrometrics ASAP2020 physisorption platform and the Brunauer–Emmett–Teller method. Samples was degassed for a period of no less than 3 h at 120 °C before measurements were made.

Samples of SHG’s (4 mL) were prepared directly in a disposable cuvette, as described in Section 2.2 below, and sealed directly after being prepared, then inserted in a Malvern Zetasizer Nano ZSP (Great Malvern, UK) model for average particle size measurements using the technique of dynamic light-scattering. The instrument was calibrated for a period of 120 s with a refractive index of water and a measurement angle of 173° back scatter at ambient temperature and set to collect readings every 600 s.

NMR spectra were recorded in 5 mm tubes on a Bruker Avance III 300 spectrometer operating at 121.495 MHz for ^31^P, using D_2_O or a D_2_O-containing capillary to provide the lock signal. NMR analyses were performed with both a 3 s pulse delay and in a gated ^1^H coupled mode in order to limit any nuclear Overhauser effects which could compromise integration measurements. In each case, 320 transients were obtained. pH measurements were made using a Jenway 350 pH meter (Spectronic Instruments, Leeds, UK) connected to an Ag/AgCl micro electrode (Aldrich) on samples in 2 mL glass vials; calibrations were performed using Buffer tablets at pH 4 (phthalate) and 7 (phosphate) supplied by Fisher.

### 2.2. Preparation of Silica Hydrogels (SHGs)

A standard operating protocol (SOP) for the production of SHGs was afforded by combining two solutions: (A) a solution of 360 μL glacial acetic acid diluted by addition of 7600 μL of ultra-pure deionized water, and (B) 1250, 2500 and 3750 μL respectively of a sodium silicate solution (≤27% SiO_2_ and ≤10% NaOH), diluted to 8000 μL with ultra-pure deionized water [61]. These compositions afford three different gels, which we identify as 0.5, 0.75 and 1.0 M silica, respectively, (the calculation is available as part of the SI). The sodium silicate solution was subsequently poured into the acid solution in a test tube. The tube was sealed and then inverted slowly (shaking the tube must be avoided) several 3–5) times to mix the components thoroughly, and then allowed to stand until gelation had been observed to take place. The gelation process that took place in the tube was confirmed by the onset of slight turbidity and inversion of the tube, which revealed the stability towards gravity of the gel (Figure 2b). SHGs were prepared using this procedure with (i) ultra-pure deionized water as outlined above, and (ii) with a salt-water substitute, standard mean ocean water (SMOW) from which the iron component had been removed [62].

### 2.3. General Methods for Preparing & Analyzing Phosphorus-Implanted SHGs

Gels were prepared as described above using the following quantities: sodium silicate solution (1200 μL), acid (400 μL either acetic or formic), ultra-distilled water or a modification of standard mean ocean water (SMOW-lite [62] which did not contain additional iron; 6600 μL). Within these basic gels, variants were prepared containing phosphorus (Pi) additives (1000 μL at 0.5 M, each component added to the acid phase of the gel system, see Table 1) and appropriate metal salts (25, 100 or 250 mg, Table 2) either layered on the gel post-preparation or dissolved in the acid phase prior to gel-phase formation. Reaction systems were left usually for a period of 72 h under ambient temperature conditions (between 20–22 °C) unless noted otherwise [Table 2 entries G1-28 (G for gel)]. For analysis, the gel was destroyed first by raising the pH upon stirring with 5 M sodium hydroxide solution (6 mL) which also resulted in precipitation of metal salts. The system was then centrifuged to compress the salts, filtered, and the water removed from the filtrate at 77 K by freeze-drying. The residue was then extracted into D_2_O (0.5 mL) prior to analysis via ^31^P-NMR spectroscopy. Any observation of pyrophosphate being present was quantified according to the percentage of total P in the extracted D_2_O. In control experiments, we confirmed that a sample of pyrophosphate [PPi(V)] remained unchanged in the presence of sodium hydroxide for a period significantly exceeding the necessary analysis time. All experiments were run in triplicate unless stated otherwise and the range of values returned indicated. Given the heterogeneous nature of the gel systems we find the spread of values to be rather larger, proportionately, than might be the case in solution experiments.

### 2.4. Control Experiments in Aqueous Solution

The following quantities were employed in the control systems:C1: H_2_O (6.6 mL), Pi(III) (0.5 M), Pi(V) (0.5 M), Fe(II) (100 mg), glacial acetic acid (200 μL)C2: H_2_O (6.6 mL), PPi(III– V) (0.5 M), Fe(II) (100 mg), glacial acetic acid (200 μL)C3: H_2_O (6.6 mL), PPi(III– V) (0.5 M), Fe(II) (100 mg), glacial acetic acid (200 μL), H_2_O_2_ (1 mL, 0.2 M)C4: H_2_O (6.6 mL), Pi(III) (0.5 M), Pi(V) (0.5 M), Fe(II) (100 mg), glacial acetic acid (200 μL), H_2_O_2_ (1 mL, 0.2 M)C5: H_2_O (6.6 mL), Pi(III) (0.5 M), Pi(V) (0.5 M), Fe(II) (100 mg), glacial acetic acid (200 μL), silica gel GF_254_ (0.2 g)C6: H_2_O (6.6 mL), PPi(III– V) (0.5 M), Fe(II) (100 mg), glacial acetic acid (200 μL), silica gel GF_254_ (0.2 g).

## 3. Results

### 3.1. Preparation and Analysis of SHGs

A hydrogel may be considered to result from the polymerization of molecular building blocks which results in a three-dimensional, macromolecular network providing some level of structural rigidity but also encapsulating significant quantities of water (Figure 2a) [63]. In the case of SHGs, this network results from silica formation either by pH modification of an alkaline solution of sodium silicate [64], or via sol-gel processing of organosilicate esters [65]. In this work we have selected to work with the former formulation mechanism as it appears to afford the most logical link to how such SHGs may have emerged within geological environments (Figure 2b) [48]. We have prepared SHGs using the method developed by Barge et al. [61], which affords hydrogels with moderate to good optical purity and gelation times that range over hours to seconds within the range of 0.5 M silica to 1.0 M silica, respectively. Critical-point drying provides samples of the silica matrix that have been subjected to scanning electron microscopy imaging. Secondary electron images of SHG matrices from 0.5 M silica are shown in Figure 3. Figure 3a reveals a low magnification image (100 µm scale bar) of a rather fractured silica surface. Under higher magnification, the formation more closely resembles branching dendritic motifs ca 5–8 µm in length [65]. The morphologies that emerge are, however, dependent upon the preparation concentrations and subsequent matrix modifications employed. However, as shown in Figure 4 for a 1.0 M silicate gel formulation, there is considerable salt formation (presumably sodium silicates, acetate and hydroxides) in addition to the silica matrix. Subsequent dialysis for a period of four days resulted in dissolution of the salts and retention of the silica matrix. Scanning electron microscopy of this, post-dialyzed, material identified both spherical [66] and alveoli silica formulations (Figure 4), the latter of which revealed a far more porous and open silica matrix than prior to dialysis (Figure 4a–d). In support of this, we find that, overall, BET surface areas of the silica matrices increase from ca 160 m^2^/g to ca 410 m^2^/g post-dialysis (see Supplementary Information) [67].

We have also monitored the pH change in a representative SHG preparation (Figure 5a; in this case with a 1.0 M silica SHG), which demonstrates the graduate rise in pH as the silicate solution is added to the acid phase, inverted gently three times and then the pH probe inserted (see Section 2.2). The graph trends towards a plateau after some 45 min wherein the pH stabilizes within the 9–10 pH range. In addition, we have also monitored the evolution of particle size distribution, as a function of time, during the gelation period of these SHGs, using dynamic light-scattering (see Appendix A for details). As such, the instrument was programmed to collect readings every 10 min for a total of 6 h. The results (Figure 5b) reveal a steady increase in particle size in solution up to a maximum of ca 450 nm after ca 100 min. Thereafter, there is a general drop-off to less than 100 nm after ca 180 min, suggesting that aggregation of larger particulates had taken place. 

### 3.2. Coupling of Pi within the Silica Hydrogel Phase

In 2008, we published an analysis of how the oxidative coupling of low oxidation state P-species, specifically phosphite (H_2_PO_3_*^−^*) and hypophosphite (H_2_PO_2_*^−^*), could take place in the presence of (i) a Fenton reactor, a combination of ferrous ions [Fe(II)] and hydrogen peroxide, or (ii) ionizing radiation [68]. In that contribution, we recognised the importance of the solution phase to achieving coupling. In the present contribution, we have sought to explore related coupling protocols in (arguably) more geologically plausible conditions than an unadulterated aqueous phase. Thus, we have examined the ability of Pi species to undergo coupling in a silica hydrogel phase in the presence of metal salts, introduced to be either homogeneous or heterogeneous to the system. The experimental work-up procedure used (Section 2.3) meant that the coupling product that we were attempting to identify was pyrophosphate, PPi(V), as all other potential coupling products [pyrophosphite, PPi(III), isohypophosphate, PPi(III– V)], would be decomposed during the high pH work-up. The basic experimental gel system is described in Section 2.3 above, but variations were made to this composition according to the variables outlined in Table 1 below. Each combination of solvent, acid type, Pi components and metal additives was examined in a series of experiments that were reviewed for their ability to generate PPi(V) by ^31^P-NMR spectroscopy.

This sequence of experiments and their outcomes are collected in Table 2. From the results of these experiments (Table 2 entries G1-27 and control experiments C1-6), we were able to make the following observations:(1)No Pi coupling was observed under any conditions where hydrochloric acid (HCl) was used as the low pH component of the gel system.(2)No metal additive, other than Fe(II), afforded any Pi coupling.(3)Successful Pi coupling protocols using acetic acid also delivered positive results for Pi coupling when acetic was replaced by formic acid.(4)No successful Pi coupling was observed when the Pi components were used seperately, Pi(III) or Pi(V).(5)Successful Pi coupling was achieved only using a 1:1 mixture of Pi(III) and Pi(V) or pure PPi(III– V).(6)Successful Pi coupling was observed both when the Fe(II) additive used was employed either in solution or as a heterogeneous addition to the pre-formed gel.(7)Successful Pi coupling was observed with Fe(II) only when formulated aerobically. No coupling was observed under anaerobic conditions.(8)Control experiments performed under aqueous (non-gelled) conditions revealed no Pi coupling but distinct oxidation of PPi(III– V) in the presence of Fe(II)-air [1.3% conversion of PPi(III– V) to PPi(V)] and pronounced oxidation in the presence of the Fenton system, Fe(II)-H_2_O_2_ [35.1% conversion of PPi(III– V) to PPi(V)]. 

## 4. Discussion

From these experiments we conclude that the following components are implicated in the formation of PPi(V): (i) a mixture of Pi(III) and Pi(V) or pure PPi(III– V); (ii) Fe(II); (iii) acetic or formic acid (not HCl); (iv) aerobic conditions or the presence of H_2_O_2_ as an oxidant; and (v) the presence of a gel system and/or a high surface area mineral. The results from Table 2 reveal that the amounts of PPi(V) produced when the feed Pi components are Pi(III) and Pi(V) (1:1 mole equivalent) are very small, within the range 0.1–0.6% (entries G1-10, Table 2) regardless of conditions such as quantity of Fe(II) used (between 25 mg–250 mg), or whether acetic or formic acid was used to gel the system (both formic and acetic acids produced PPi(V) with the same order of magnitude). However, significantly, we find that when PPi(III– V) is used instead of Pi(III) and Pi(V), then PPi(V) was observed at 2–3% (±1) when performed under aerobic conditions (see Figure 6 for an exemplar ^31^P-NMR spectrum of the product from reaction G14) which was still a small value overall, but nevertheless 6–10 times greater than was observed with the individual Pi components themselves. When H_2_O_2_ was substituted for air, it was observed that PPi(III– V) was converted to PPi(V) far more efficiently, as expected under such Fenton-like conditions (entries G13-18, Appendix A, Table 2 [67]); even removing the Fe(II) and employing H_2_O_2_ alone resulted in conversion of PPi(III– V) to PPi(V) on the order of 1–2% (entries G19-21). A series of control experiments was also performed (C1-C6; Table 2) under non-gelled, aqueous conditions, which revealed oxidation of PPi(III– V) to PPi(V) at ca 1–2% under aerobic conditions (C2 & C6); a considerable jump then to *ca* 30% oxidation in the presence of H_2_O_2_ is observed (C3) and there is no evidence of Pi coupling when a mixture of Pi(III) and Pi(V) is used (C1, C4 & C5). 

On the basis of the data presented, therefore, we envisage a process within the presence of the silica hydrogel in which Pi(III) and Pi(V) undergo a condensation to afford PPi(III– V) as an intermediary phase. This then undergoes Fe(II)-mediated oxidation via a Fenton-like reaction in a manner analogous to that reported by Pasek and Kee, in the aqueous phase [68]. That we do not see any intermediate PPi(III– V) in our experiments is, we believe, a consequence of the high pH work-up procedure, which leads to rapid hydrolysis of isohypophosphate [69] (pyrophosphate is considerably more stable under such conditions). We speculate on two mechanistic possibilities for the primary in-gel Pi coupling. In one possibility, an Fe(II)-O_2_(air) mediated process leads to Pi radicals which combine to afford PPi(V) (Scheme 1a). In the second possibility, a reaction between acetic acid and Pi to afford acetylphosphorus species which subsequently engage in condensation to afford PPi(III– V), which then is oxidised by the Fe(II)-oxidant system to PPi(V) (Scheme 1b). Any oxidative pathway, such as that outlined in Scheme 1a, would most likely be mechanistically complex, as has already been demonstrated for the solution and mineral-supported Fe(II)-O_2_(air) redox system [70]. As we do not observe Pi coupling in the aqueous, non-gel phase in the presence of Fe(II)-O_2_(air) alone, any such oxidative coupling may then be a consequence of the environment changing from the aqueous to salt-gel phase; further studies are needed to explore this possibility. In the second alternative, outlined in Scheme 1b, an acetylphosphorus intermediate (**1**) is formed by reaction between Pi species and acetic acid, which subsequently undergoes condensation with Pi to afford PPi(III), PPi(III– V) or PPi(V), a process for which precedent exists [56]. Given that such a condensation between Pi and acetic acid would be significantly less likely in a high rather than low pH environment, we speculate that any formation of acetylphosphorus species is more likely to take place prior to the addition of silicate solution and hence gel formation. Furthermore, since our experiments seem to suggest that Pi coupling does not take place in the aqueous phase and that coupling within the gel requires also the presence of Pi(III), one might speculate that the presence of a silica matrix of high surface area within the high-pH (ca 9–10 pH units) gel phase is having some influence over the coupling process. The nature of such an effect would require further, more detailed mechanistic investigations which are currently on-going.

## 5. Conclusions

We have attempted here to demonstrate that metal-mediated coupling of phosphorus oxyacids (Pi) can occur under ambient temperature conditions, without additional condensation? Agents in environments that may be considered more geologically relevant than pure aqueous solutions. In so doing, we chose to examine such processes in geologically-relevant silica hydrogels [46,47,48], which have precedent as a substitute for a putative primitive cellular cytoplasm [31,32,33,34,35]. We find that the coupling of mixtures of Pi(III) and Pi(V) does take place, albeit in low overall conversions in these preliminary static experiments, leading to the observation of PPi(V). Interestingly, we observe that no such coupling takes place under the same conditions in aqueous (non-gel) control experiments. We do not yet have sufficient evidence that would allow us to propose a definitive mechanism for such coupling, but the weight of evidence to date leads us to favour a Pi-coupling process to afford PPi(III– V) combined with an Fe(II)-air mediated oxidation of this oxyacid ultimately to PPi(V). 

Experiments to explore the mechanism of coupling are on-going as well as a wider examination of the effects of the silica hydrogel environment on other processes of relevance to prebiotic chemistry.

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
