# Peer review of "Chemical Transformations in Proto-Cytoplasmic Media. Phosphorus Coupling in the Silica Hydrogel Phase"

_life, 2017, doi:10.3390/life7040045_

Round 1

Reviewer 1 Report

I think this is an interesting study addressing a critical question about the origin of life; how did phosphorus chemistry become incorporated into proto-biochemistry. This paper examines the effect of silica gel on coupling phosphite to phosphate, as a model for a proposed gel-like environment for abiogeneiss and a model for early phosphate metabolism respectively.

The authors have carried out a series of exploratory experiments looking at the effect of a silica gel on the condensation of phosphate with phosphite. They define a reproducible gel making method, and characterise the gel physically as a structure with pore sizes comparable to the size of a protein, and hence similar in some respects to the ‘crowded’ environment inside a cell. They find that, in this milieu, some small amount of phosphorus(V) ester/anhydride is formed.

The results are rather telegraphic; the important results are in lines 216 – 249, which are a qualitative table and some text. How much PPi(V)? Maybe an example NMR scan (for those who would like to see such data)? I know this is in the SI, but to have to dig down into the SI to see critical data disrupts the flow. Did acetic and formic acid give the same yield?  Point 6 is very under-explained – what sort of gel? Fine suspension? Huge lumps?

The results show that PPI(II/V) is oxidized to PPi(V) by Fe and O2 or H2O2 (not that surprising), and Pi(II)+Pi(V) forms PPi(V) at low efficiency (more surprising). It is not clear to me whether they conducted control experiments with colloidal silica, or whether the ‘aqueous’ controls are without silica present, i.e. is this an effect of the gel structure or of a high surface area of hydrated silica? Given the occurrence of colloidal silica in hydrothermal systems, this is pretty important.

The mechanistic discussion I find hard to follow and not than convincing. The idea that the silica is decorated with acetate (lines 294-6) at pH 9 highly unlikely. Stapelton et al, cited a precedence for this, discuss silica exposed to anhydrous acetic acid and acetyl chloride vapour, not aqueous phase reaction. Si-O-C(=O) bonds are notoriously liable to hydrolysis. See (as example) the MDS for ACETOXYTRIMETHYLSILANE at Gelest, which explicitly states sensitivity to moisture. In general the paper tends to have long stretches of text that I find hard to follow. Simpler sentences and shorter paragraphs would be easier to understand. 

The pH and concentration calculations seem odd. The authors state the concentration of acetic acid is 7mM (line 308). If I add the volumes up properly in 2.3, the gel is ~10mls, so that is 400ul of acetic acid in 10 mls, ~0.4g/60 daltons/0.01litres = 400mM (density of glacial acetic acid ~1 g/ml – the paper states that acetic and formic acid were glacial – line 89).

I also feel very uneasy about a lot of the language here. The conclusion starts with the sentence “We have attempted here to demonstrate that metalmediated coupling of phosphorus oxyacids (Pi) can occur under ambient temperature conditions, without additional condensing agents under environments which may be considered more geologically relevant that pure aqueous solutions.” That is fine, it is what the authors did. But there is  a weight of speculation in other parts that is not relevant to or supported by the data, and in some cases I think is mistaken.

This paper takes as its starting point the concept that life originated in, or as, a hydrogel, citing Trevors and Pollack. It is not clear to me that some of the statements made in this paper about the T&P hypothesis are entirely correct. Thus a hydrogel can maintain an ion concentration differential between inside and outside, but this is a thermodynamic minimum, not a dynamically maintained, out-of-equilibrium system. Hydrogels can be stable in contact with water, but the gel-like interior of the cell (with which these authors draw comparison) obviously is not. As Trevors Gerald and Pollack themselves comment (2005), hydrogels exclude solutes, and so embedding large solutes in a hydrogel is a bit of a problem. This is not central to their paper, and I think the general idea that a) life is a gel, not a solution and b) hydogels are interesting potential milieus for OOL both are relevant; the first is obviously true and the second interesting. But personally I would dial back the proselytising for T&P a bit in the introduction, just say that they suggest a very interesting alternative scenario to the ‘warm little pond’, and the purpose of this paper  is to explore some chemical aspects of a specific gel.

The authors also imply (lines 51-54) that the idea that molecular crowding in the cell is derived from T&P’s work, and cite a few papers from 2014-7 to support this. This is quite untrue. Google Scholar gives over 10,000 references to the search [“molecular crowding” cell] . This is a very well recognised concept. See for example [Minton, Allen P. "Excluded volume as a determinant of macromolecular structure and reactivity." Biopolymers 20.10 (1981): 2093-2120], [Zimmerman, Steven B., and Allen P. Minton. "Macromolecular crowding: biochemical, biophysical, and physiological consequences." Annual review of biophysics and biomolecular structure 22.1 (1993): 27-65], [Ellis, R. John. "Macromolecular crowding: an important but neglected aspect of the intracellular environment." Current opinion in structural biology 11.1 (2001): 114-119.] and many others. T&P have suggested that hydrogels provide a prebiotic analogue of this intracellular milieu, which is fair enough if, in my view, implausible. But crowding is an old and established concept.

The existence of silica gel in the real world is argued from three papers. Grenne and Slack is relevant, reporting a proposed fall-out from a hydrothermal plume of highly hydrated gel.  Interstingly, I cannot find *any* examples of where such a gel has been observed in a modern hydrothermal system, though. Suggests it is not common. Kirkpatrick et al show morphological examples of a hydrated amorphous silica phase, and cite two papers relating the similar morphology in the lab – both discuss quite dense phases of silica more akin to water-rich quartz than the material Gorrell et al describe. Papineau has one passing reference to gels, and is not really relevant.  In reality though silica is precipitated in hydrothermal systems as colloidal particles (see  eg Ohsawa, Shinji, et al. Journal of Volcanology and Geothermal Research 113.1 (2002): 49-60. ), which is not the same as a gel. A bit more support for the idea that their gel (which from their description only forms is it is not shaken too much during the gelation phase, and contains <10% silica by weight) is even distantly like something that could form geochemically would be good, or a clear argument why this does not matter.

The authors pursue their long-standing agenda that phosphate was a significant player in prebiotic phosphorus chemistry. Fair enough, but I think for the unwary reader they should flag that this is one hypothesis about how phosphorus came to incorporated into the metabolism that evolved into LUCA, and that no modern organism uses phosphite for energy transduction (or for anything else? My knowledge runs out here, microorganisms are astonishingly adaptable, but as far as I know there are no metabolites with a P-H bond known apart from those involved directly in metabolising exogenously provided phosphite to phosphate.)

The terminology of ‘wild type’ and ‘mutants’ in respect to their gels is just irritating (lines 137 – 143). This might be amusing lab. banter, but is not appropriate for a paper. They are baseline and variant gels, or undoped and doped I suppose.

The system was then centrifuged to compress the salts, filtered and the water removed from the filtrate at 77 K.”(line 146-7). Does this mean they were freeze-dried?

I am sure the spreadsheet snapshot Table 2 could be presented in a simpler way. What do the different colours signify? Is the green PPi(V) column at the right an input or an output of the experiment? We find this later on (lines 252-5), but a more informative heading might help comprehension.

 “We have also monitored the pH change in a representative SHG preparation (Figure 4a; in this case  with a 1.0 M silica SHG), which demonstrates the graduate rise in pH as the silicate solution is added slowly to the acid phase (see section 2.2).(lines 201-3). This is not consistent with the methods section, which says the two solutions are mixed and gently shaken, not mixed over a 45 minute period. This must rather reflect slow chemistry going on. The next few sentences in this paragraph on particle size also do not make sense in light (sorry) of Figs 2b and 3 which suggest a continuous material. So are these a suspension or particles or a continuous material, and if the latter are you measuring the size of average density fluctuations rather than of particles?

Author Response

Reviewer #2

I think this is an interesting study addressing a critical question about the origin of life; how did phosphorus chemistry become incorporated into proto-biochemistry. This paper examines the effect of silica gel on coupling phosphite to phosphate, as a model for a proposed gel-like environment for abiogeneiss and a model for early phosphate metabolism respectively.

The authors have carried out a series of exploratory experiments looking at the effect of a silica gel on the condensation of phosphate with phosphite. They define a reproducible gel making method, and characterise the gel physically as a structure with pore sizes comparable to the size of a protein, and hence similar in some respects to the ‘crowded’ environment inside a cell. They find that, in this milieu, some small amount of phosphorus(V) ester/anhydride is formed.

The results are rather telegraphic; the important results are in lines 216 – 249, which are a qualitative table and some text. How much PPi(V)? Maybe an example NMR scan (for those who would like to see such data)? I know this is in the SI, but to have to dig down into the SI to see critical data disrupts the flow. Did acetic and formic acid give the same yield?  Point 6 is very under-explained – what sort of gel? Fine suspension? Huge lumps?

The Reviewer has some good points here. We have included an image of a 31P-NMR spectrum in the manuscript proper. We have offered some further clarification of point 6 somewhat in the text and also included a statement on acetic/formic acid efficacy which essentially puts the effects of both at the same of of magnitude in generating PPi(V). Beyond this, we do not such fine-grained data to be able to compare in detail. The narrative changes have been indicated by the descriptor R2.1 (Reviewer 2.change 1)

The results show that PPI(II/V) is oxidized to PPi(V) by Fe and O2 or H2O2 (not that surprising), and Pi(II)+Pi(V) forms PPi(V) at low efficiency (more surprising). It is not clear to me whether they conducted control experiments with colloidal silica, or whether the ‘aqueous’ controls are without silica present, i.e. is this an effect of the gel structure or of a high surface area of hydrated silica? Given the occurrence of colloidal silica in hydrothermal systems, this is pretty important.

 This again is a good point. The control that we did was with aqueous solution with no gel or silica present. Thus, we cannot say for sure that the reason for the coupling is connected to an effect of the gel phase or high surface area. We have included that clarifying point. The narrative changes have been indicated by the descriptor R2.2.

The mechanistic discussion I find hard to follow and not than convincing. The idea that the silica is decorated with acetate (lines 294-6) at pH 9 highly unlikely. Stapelton et al, cited a precedence for this, discuss silica exposed to anhydrous acetic acid and acetyl chloride vapour, not aqueous phase reaction. Si-O-C(=O) bonds are notoriously liable to hydrolysis. See (as example) the MDS for ACETOXYTRIMETHYLSILANE at Gelest, which explicitly states sensitivity to moisture. In general the paper tends to have long stretches of text that I find hard to follow. Simpler sentences and shorter paragraphs would be easier to understand. 

Valid points indeed. I have reviewed my proposed mechanism in line with the reviewer’s comments which have been altered (descriptor R2.3). The reviewer makes a good point about the pH dependency of acetoxytrimethylsilane hydrolysis. Moreover, given that the Pi components are incubated within the acid phase prior to introduction of the silicate and hence gel formation (section 2.3) it may seem more logical to consider a Pi-acetic acid interaction phase prior to gel formation. I have amended the discussion accordingly which also results in more discursive speculative elements being redundant and hence removed (descriptor R2.4)

The pH and concentration calculations seem odd. The authors state the concentration of acetic acid is 7mM (line 308). If I add the volumes up properly in 2.3, the gel is ~10mls, so that is 400ul of acetic acid in 10 mls, ~0.4g/60 daltons/0.01litres = 400mM (density of glacial acetic acid ~1 g/ml – the paper states that acetic and formic acid were glacial – line 89).

Many thanks to the reviewer for correcting this error. I have now made the necessary changes on the basis that 400 mL acetic acid (RMM 60.05; d = 1.048 @20oC) in a total volume of 9.9 mL affords (1.048 x 0.4)/60.05 = 6.98 mmoles of acetic acid in the gel volume; a molarity of therefore 0.7 M [(6.98 x 10-3 x 1000)/9.9]. This is to be compared with a total Pi molarity of 1M. This corrected calculation would imply that the concentration differential between acetic acid and Pi components to be far less than pointed out in the paper, providing a stronger rather than weaker argument for the acid component being significant in more than just a pH manner. I have removed this calculation from the narrative as it now adds relatively little (descriptor R2.5)

I also feel very uneasy about a lot of the language here. The conclusion starts with the sentence “We have attempted here to demonstrate that metalmediated coupling of phosphorus oxyacids (Pi) can occur under ambient temperature conditions, without additional condensing agents under environments which may be considered more geologically relevant that pure aqueous solutions.” That is fine, it is what the authors did. But there is  a weight of speculation in other parts that is not relevant to or supported by the data, and in some cases I think is mistaken.

This paper takes as its starting point the concept that life originated in, or as, a hydrogel, citing Trevors and Pollack. It is not clear to me that some of the statements made in this paper about the T&P hypothesis are entirely correct. Thus a hydrogel can maintain an ion concentration differential between inside and outside, but this is a thermodynamic minimum, not a dynamically maintained, out-of-equilibrium system. Hydrogels can be stable in contact with water, but the gel-like interior of the cell (with which these authors draw comparison) obviously is not. As Trevors Gerald and Pollack themselves comment (2005), hydrogels exclude solutes, and so embedding large solutes in a hydrogel is a bit of a problem. This is not central to their paper, and I think the general idea that a) life is a gel, not a solution and b) hydogels are interesting potential milieus for OOL both are relevant; the first is obviously true and the second interesting. But personally I would dial back the proselytising for T&P a bit in the introduction, just say that they suggest a very interesting alternative scenario to the ‘warm little pond’, and the purpose of this paper  is to explore some chemical aspects of a specific gel.

Cautionary note taken and accepted. The T&P sections have been altered accordingly (descriptor R2.6).

The authors also imply (lines 51-54) that the idea that molecular crowding in the cell is derived from T&P’s work, and cite a few papers from 2014-7 to support this. This is quite untrue. Google Scholar gives over 10,000 references to the search [“molecular crowding” cell] . This is a very well recognised concept. See for example [Minton, Allen P. "Excluded volume as a determinant of macromolecular structure and reactivity." Biopolymers 20.10 (1981): 2093-2120], [Zimmerman, Steven B., and Allen P. Minton. "Macromolecular crowding: biochemical, biophysical, and physiological consequences." Annual review of biophysics and biomolecular structure 22.1 (1993): 27-65], [Ellis, R. John. "Macromolecular crowding: an important but neglected aspect of the intracellular environment." Current opinion in structural biology 11.1 (2001): 114-119.] and many others. T&P have suggested that hydrogels provide a prebiotic analogue of this intracellular milieu, which is fair enough if, in my view, implausible. But crowding is an old and established concept.

This is fair enough and indeed, as the reviewer points out, molecular crowding is a concept that has been around for some time and certainly one not “invented” by T&P; I most certainly did not wish to convey this! I’ve looked carefully at the narrative now and hope to have clarified this (descriptor R2.7)

The existence of silica gel in the real world is argued from three papers. Grenne and Slack is relevant, reporting a proposed fall-out from a hydrothermal plume of highly hydrated gel.  Interstingly, I cannot find *any* examples of where such a gel has been observed in a modern hydrothermal system, though. Suggests it is not common. Kirkpatrick et al show morphological examples of a hydrated amorphous silica phase, and cite two papers relating the similar morphology in the lab – both discuss quite dense phases of silica more akin to water-rich quartz than the material Gorrell et al describe. Papineau has one passing reference to gels, and is not really relevant.  In reality though silica is precipitated in hydrothermal systems as colloidal particles (see  eg Ohsawa, Shinji, et al. Journal of Volcanology and Geothermal Research 113.1 (2002): 49-60. ), which is not the same as a gel. A bit more support for the idea that their gel (which from their description only forms is it is not shaken too much during the gelation phase, and contains <10% silica by weight) is even distantly like something that could form geochemically would be good, or a clear argument why this does not matter.

Yes, the reviewer raises a valid point here, The actual gel recipe that we are using here is not an especially good match to any geological silica gel phase that I am aware of. However, the thrust of what we are attempting to demonstrate here is not to mimic a geological environment but to highlight how certain chemistries may change when the phase is altered. I have made this clearer in the text (descriptor R2.8)

The authors pursue their long-standing agenda that phosphate was a significant player in prebiotic phosphorus chemistry. Fair enough, but I think for the unwary reader they should flag that this is one hypothesis about how phosphorus came to incorporated into the metabolism that evolved into LUCA, and that no modern organism uses phosphite for energy transduction (or for anything else? My knowledge runs out here, microorganisms are astonishingly adaptable, but as far as I know there are no metabolites with a P-H bond known apart from those involved directly in metabolising exogenously provided phosphite to phosphate.)

This is a fair point and I’ve changed the text accordingly to make this clearer (descriptor R2.9)

The terminology of ‘wild type’ and ‘mutants’ in respect to their gels is just irritating (lines 137 – 143). This might be amusing lab. banter, but is not appropriate for a paper. They are baseline and variant gels, or undoped and doped I suppose.

Accepted and changed in the narrative (descriptor R2.10)

The system was then centrifuged to compress the salts, filtered and the water removed from the filtrate at 77 K.”(line 146-7). Does this mean they were freeze-dried?

Yes, the samples were lyophilized. I’ve added that point (descriptor R2.11)

I am sure the spreadsheet snapshot Table 2 could be presented in a simpler way. What do the different colours signify? Is the green PPi(V) column at the right an input or an output of the experiment? We find this later on (lines 252-5), but a more informative heading might help comprehension.

Yes, I accept this point also; it is made by reviewer #3 as well. The use of color is just as a binary differentiation here as to which elements in each columns are relevant for each experimental entry. I thought that the use of color would help guide the eye but, being color-blind myself, I accept that this may not have been a good choice for the majority of readers. I have changed this now to text in an effort to avoid confusion (descriptor R2./R3.3).

 “We have also monitored the pH change in a representative SHG preparation (Figure 4a; in this case with a 1.0 M silica SHG), which demonstrates the graduate rise in pH as the silicate solution is added slowly to the acid phase (see section 2.2).”(lines 201-3). This is not consistent with the methods section, which says the two solutions are mixed and gently shaken, not mixed over a 45 minute period. This must rather reflect slow chemistry going on. The next few sentences in this paragraph on particle size also do not make sense in light (sorry) of Figs 2b and 3 which suggest a continuous material. So are these a suspension or particles or a continuous material, and if the latter are you measuring the size of average density fluctuations rather than of particles?

Indeed the two solutions are mixed and gently inverted several times prior to insertion of the pH probe. Essentially, we are looking here at the change in bulk pH over a 45 min period as gelation is taking place. I have clarified that now in the text (descriptor R2.13). Thanks for pointing this out.

As for the dynamic light scattering data my interpretation of what we are seeing in solution is gradual increase in particle size average vs time followed by silica matrix formation and removal from the sol phase. The silica matrix displayed in the SEM images does indeed show continuous material but this is of course after significant processing via critical point drying on mature (several day-old) hydrogel. I would envisage that DLS and SEM are offering us a snap-shot of what is present in the sol and solid phases respectively. 

Reviewer 2 Report

This manuscript details a study into the prebiotic formation of [PPi(V)] from other mono-phosphorous containing species in the presence of a silica hydrogel. Hydrogels are a potential protocell mimic but have not received much attention to date. Those which have been studied tend to be hydrogels formed from organic molecules. The novelty of this work is the use of silica for form an inorganic hydrogel, which may mimic some early geological conditions. The authors found that when the reaction was carried out in solution no [PPi(V)] was formed, but when it was carried out in the silica hydrogel quantities of [PPi(V)] were formed. While these quantities are not large <5% they are significant as a proof of principle. This work will be of interest to those studying the processes at the dawn of life and may encourage others to investigate hydrogels as protocellular environments.

Author Response

Reviewer #1

 This manuscript details a study into the prebiotic formation of [PPi(V)] from other mono-phosphorous containing species in the presence of a silica hydrogel. Hydrogels are a potential protocell mimic but have not received much attention to date. Those which have been studied tend to be hydrogels formed from organic molecules. The novelty of this work is the use of silica for form an inorganic hydrogel, which may mimic some early geological conditions. The authors found that when the reaction was carried out in solution no [PPi(V)] was formed, but when it was carried out in the silica hydrogel quantities of [PPi(V)] were formed. While these quantities are not large <5% they are significant as a proof of principle. This work will be of interest to those studying the processes at the dawn of life and may encourage others to investigate hydrogels as protocellular environments.

 Authors response

The authors thank the reviewer for their summary and for recognizing that geological gels are a novel environment in which to perform prebiotic chemistry experiments.

Reviewer 3 Report

The authors have correctly recognized a certain lacuna as far as the origin-of-life research goes pertaining to experiments performed in a gel phase.

However, some research in this direction was done before and not all of them present as enthusiastic results as  the Authors hope for in the current thesis.

Particularly recent research from the Szostak Lab have shown not encouraging results as far as retention of RNA monomers in poly-Lysine gel for prebiotic RNA oligomerization purposes. [Jia, T. Z., Hentrich, C., & Szostak, J. W. (2014). Rapid RNA exchange in aqueous two-phase system and coacervate droplets. Origins of Life and Evolution of Biospheres, 44(1), 1-12.]

This contrary view on the gel approach should be somehow discussed by authors in the paper.

Also Christine D. Keating from Pennsylvania State University has published a number of origin-of-life related research on gels inside liposomes. It would be interesting to know the authors opinion on that line of investigation and how it relates to their own experiments. Unless the authors believe that such discussion would stray to far from the main focus of the paper.

In line 140 the authors use the word “mutants”, which in my opinion is not appriopriate in this context. A simple “variantions” will suffice.

The biggest current problem of the paper is Table 2. For some reason it is placed on the fifth page, several pages before Table 1 and several pages before it is being discussed in the text. Also the table itself is not particularly readeble or user friendly. The table has lots of colours but they are not explained in the legend. What do they denote? Is it a simple binary and the different colours simply denote different column? It is not clear at this stage. I strongly urge the authors to reconsider its presentation and come up with something more easy understandable.

Author Response

Reviewer #3

 The authors have correctly recognized a certain lacuna as far as the origin-of-life research goes pertaining to experiments performed in a gel phase.

However, some research in this direction was done before and not all of them present as enthusiastic results as  the Authors hope for in the current thesis.

Particularly recent research from the Szostak Lab have shown not encouraging results as far as retention of RNA monomers in poly-Lysine gel for prebiotic RNA oligomerization purposes. [Jia, T. Z., Hentrich, C., & Szostak, J. W. (2014). Rapid RNA exchange in aqueous two-phase system and coacervate droplets. Origins of Life and Evolution of Biospheres, 44(1), 1-12.]

This contrary view on the gel approach should be somehow discussed by authors in the paper.

 This is a good point and I have included some qualifying remarks to this effect including reference to Szostak’s work (descriptor R3.1)

Also Christine D. Keating from Pennsylvania State University has published a number of origin-of-life related research on gels inside liposomes. It would be interesting to know the authors opinion on that line of investigation and how it relates to their own experiments. Unless the authors believe that such discussion would stray to far from the main focus of the paper.

In line 140 the authors use the word “mutants”, which in my opinion is not appriopriate in this context. A simple “variantions” will suffice.

 Yes, agreed. This has been changed as descriptor R2.10 (see above)

The biggest current problem of the paper is Table 2. For some reason it is placed on the fifth page, several pages before Table 1 and several pages before it is being discussed in the text. Also the table itself is not particularly readeble or user friendly. The table has lots of colours but they are not explained in the legend. What do they denote? Is it a simple binary and the different colours simply denote different column? It is not clear at this stage. I strongly urge the authors to reconsider its presentation and come up with something more easy understandable.

 Yes, I accept this point also; it is made by reviewer #2 as well. The use of color is just as a binary differentiation here as to which elements in each columns are relevant for each experimental entry. I thought that the use of color would help guide the eye but, being color-blind myself, I accept that this may not have been a good choice for the majority of readers. I have changed this now to text in an effort to avoid confusion (descriptor R2./R3.3). I do not know why the table appears on page 5 as it does not do so in my original version, where it appears at the end. I presume that this is a formatting issue which can easily be corrected?